# Maternal High-Fat Diet Reduces Type-2 Neural Stem Cells and Promotes Premature Neuronal Differentiation during Early Postnatal Development

**DOI:** 10.3390/nu14142813

**Published:** 2022-07-08

**Authors:** Xiaoxuan Hu, Jing An, Qian Ge, Meiqi Sun, Zixuan Zhang, Zhenlu Cai, Ruolan Tan, Tianyou Ma, Haixia Lu

**Affiliations:** 1Department/Institute of Neurobiology, School of Basic Medical Sciences, Xi’an Jiaotong University Health Science Center, Xi’an 710061, China; xiaoxuanhu21@163.com (X.H.); jingan@mail.xjtu.edu.cn (J.A.); geqian2020@gmail.com (Q.G.); smqmina44@163.com (M.S.); zhangzixuan_dsa@163.com (Z.Z.); 15339051098@163.com (Z.C.); tanruolan19930117@163.com (R.T.); 2Key Laboratory of Ministry of Education for Environment and Genes Related to Diseases, Xi’an Jiaotong University Health Science Center, Xi’an 710061, China; 3Department of Human Anatomy & Histoembryology, School of Basic Medical Sciences, Xi’an Jiaotong University Health Science Center, Xi’an 710061, China; 4School of Public Health, Xi’an Jiaotong University Health Science Center, Xi’an 710061, China

**Keywords:** maternal high-fat diet, dentate gyrus, neural stem cells behavior, early postnatal period, neuronal differentiation

## Abstract

Maternal obesity or exposure to a high-fat diet (HFD) has an irreversible impact on the structural and functional development of offspring brains. This study aimed to investigate whether maternal HFD during pregnancy and lactation impairs dentate gyrus (DG) neurogenesis in offspring by altering neural stem cells (NSCs) behaviors. Pregnant Sprague-Dawley rats were fed a chow diet (CHD) or HFD (60% fat) during gestation and lactation. Pups were collected on postnatal day 1 (PND 1), PND 10 and PND 21. Changes in offspring body weight, brain structure and granular cell layer (GCL) thickness in the hippocampus were analyzed. Hippocampal NSCs behaviors, in terms of proliferation and differentiation, were investigated after immunohistochemical staining with Nestin, Ki67, SOX2, Doublecortin (DCX) and NeuN. Maternal HFD accelerated body weight gain and brain structural development in offspring after birth. It also reduced the number of NSCs and their proliferation, leading to a decrease in NSCs pool size. Furthermore, maternal HFD intensified NSCs depletion and promoted neuronal differentiation in the early postnatal development period. These findings suggest that maternal HFD intake significantly reduced the amount and capability of NSCs via reducing type–2 NSCs and promoting premature neuronal differentiation during postnatal hippocampal development.

## 1. Introduction

Maternal nutritional changes are particularly important in hippocampal dentate gyrus (DG) neurogenesis in offspring during the developmental window [1]. Evidence has shown that the risk of neurodevelopmental disorders in offspring increases dramatically with a rise in maternal obesity [2,3,4,5]. Maternal high-fat diet (HFD) during pregnancy and lactation contributed to the increase in risk of behavioral and mental health disorders in offspring [6].

As one of the few regions with continuous neurogenesis in the adult brain, DG development primarily occurs postnatally [7,8]. It forms the basis of adult learning memory and cognitive performance [9,10]. Neural stem cell (NSC) is a group of self-renewing cells that undergo a protracted dividing, differentiation and depletion during early development to ensure lifelong maintenance of the hippocampal NSCs population and potential [11,12,13]. The balance between self-renewal and differentiation is achieved through the control of cell division. DG neurogenesis originates from a population of radial glia-like precursor cells (type–1 NSCs) that have astrocytic properties and are marked by glial fibrillary acidic protein (GFAP) [14]. They give rise to intermediate progenitor cells (type–2a NSCs) and then to neuronal (type–2b) phenotypes via asymmetric division. Through a migratory neuroblast-like stage (type–3a), they generate into mature neurons (type–3b) [10]. A previous study has indicated that failure to control cell division properly may lead to premature depletion of NSCs pool or abnormal growth and impaired differentiation [15].

Neurogenesis is a multistage process that is influenced by many variables [16]. Stimuli that affect neurogenesis may target distinct aspects of this process [17]. NSCs are sensitive to metabolic fluctuations [18]. Previous studies have indicated that maternal diet-induced obesity leads to peroxidized lipid accumulation and impairment of hippocampal neurogenesis during the early life of offspring [19]. Meanwhile, some in vitro data also indicated that the proliferation and differentiation of offspring DG neural progenitors decreased under the influence of maternal HFD prior to and during pregnancy [20]. The offspring in adulthood exhibited decreased neurogenesis, apoptosis and neuronal differentiation [21,22,23] and impaired synaptic plasticity and memory [24,25].

Previous studies mainly focus on the effects of maternal HFD on adult hippocampal neurogenesis and cognitive disorders in adolescents or young adult offspring. However, its effect on hippocampal neurogenesis during the early postnatal stages is not fully understood. In this study, we investigated how excessive fat intake in pregnant female rats affects NSCs biological behaviors during the early DG development of offspring.

## 2. Materials and Methods

### 2.1. Animal Model of HFD

All animal procedures were approved by the Institutional Animal Care and Use Committee of the Xi’an Jiaotong University. Maternal HFD animal model was constructed following a standard protocol [26]. Virgin female (230–250 g) and male (270–300 g) Sprague–Dawley rats were obtained from the Experimental Animal Center of Xi’an Jiaotong University. Before and during mating, all rats were fed with a chow diet (CHD, standard chow diet) and housed in polypropylene cages in a 12 h light/dark cycle room at constant temperature (24 ± 2 °C) and relative humidity (60~80%).

Two females and one male rat were housed in a cage overnight to mate, and pregnancy was confirmed by examining vaginal smears for the presence of sperm, designated as the day of conception and day 0 of gestation. Pregnant rats were divided into 2 groups according to their diet (Figure 1A): CHD (lab diet, 5001, 3.00 kcal/g, containing 58.6% carbohydrate, 28.4% protein, and 13.5% fat by calorie; Beijing Ke Ao Xie Li, Beijing, China; *n* = 9) and HFD (research diet D12492, 4.30 kcal/g, containing 20% carbohydrate, 20% protein, and 60% fat by calorie; New Brunswick, NJ, USA; *n* = 9). All dams remained on their diet throughout pregnancy and lactation. Dams were weighed on gestation days (GD) 1, GD 10, and GD 20. The day of parturition was postnatal day 0 (PND 0). Litters were adjusted to 10 pups without gender selectivity for each dam at PND 1 to ensure standardized nutrition during suckling.

### 2.2. Measurement of Offspring Brain Structure and Neocortical Thickness

Each pup was weighed on PND 1, PND 10 and PND 21 in both groups. Brain wet weight, biparietal diameter (BPD, the widest distance of the head between the left and right sides), and occipitofrontal diameter (OFD, the distance of the head from front to back) were measured to assess brain development. Neocortical thickness was measured on PND 1. Cortical thickness was measured at 20% (dorsal), 50% (mediolateral) and 80% (lateral) distances from the dorsal midline according to a previous procedure [27] (Figure 1I). Other measurements were performed after brain tissue collection.

### 2.3. Tissue Collection

All experiments were conducted without sex discrimination. On PND 1, PND 10 and PND 21, for immunohistochemical staining and analysis, 3 to 4 pups per litter were perfused via the ascending aorta with 0.01 M phosphate-buffered saline (PBS, pH 7.4), followed by 4% paraformaldehyde (PFA) under overdose of anesthesia (0.75% sodium pentobarbital, 7.5 mg/kg, i.p.). The whole brain was removed from the cranium and post-fixed in PFA overnight at 4 °C. Brains were then transferred to 30% sucrose for cryoprotection and sectioned into slices at 15 μm. For Western blot analysis, 6 to 7 pups per litter were euthanized. The whole hippocampus was rapidly dissected and kept at −80 °C after frozen in liquid nitrogen.

### 2.4. Immunohistochemical Staining and Cell Counting

Immunostaining was performed following the standard protocol and optimized in our laboratory [28]. Briefly, 4 to 6 brain sections from each pup were collected and washed with 0.01 M PBS for 15 min. Following permeabilization with 0.3% TritonX-100, the sections were washed and blocked with 5% bovine serum albumin (BSA, Sigma, St. Louis, MO, USA) for 2 h at room temperature (RT). Subsequently, 100 μL primary antibodies was added into each section and kept in a humid chamber overnight at 4 °C. On the second day, the brain sections were washed with PBS and then incubated with fluorescence conjugated secondary antibodies for 2 h at RT in dark. Finally, the sections were counterstained with 0.1 µg/mL 4’,6-diamidino-2-phenylindole (DAPI, Sigma, St. Louis, Mo, USA) for 5 min and rinsed with PBS, coverslipped with DAKO fluorescent medium (Sigma, St. Louis, Mo, USA). Stained sections were observed and imaged using a fluorescence microscope (BX57; Olympus Corporation, Tokyo, Japan).

Antibodies were used as follows: primary antibodies, including anti-Nestin (1:400, mouse-monoclonal; Novus Biologicals, Colorado, USA), anti-Sox2 (1:300, mouse-monoclonal; Abcam, Cambridge, MA, USA), anti-NeuN (1:300, mouse-monoclonal; Novus Biologicals, Colorado, USA), anti-GFAP (1:500, rabbit polyclonal; Abcam, Cambridge, MA, USA), anti-Ki67 (1:400, Sigma, St. Louis, Mo, USA), and anti-doublecortin (DCX, 1:500, rabbit polyclonal; Cell Signaling Technology, Danvers, MA, USA); secondary antibodies, including Alexa Fluor 488 conjugated donkey anti-mouse IgG (H+L), Alexa Fluor 594 conjugated donkey anti-mouse IgG, Alexa Fluor 488 donkey anti-rabbit IgG (H+L), and Alexa Fluor 594 conjugated donkey anti-rabbit IgG (1:500, Invitrogen; Thermo Fisher Scientific, Waltham, MA, USA).

The hippocampal dentate gyrus regions were selected for imaging, and three visual fields per animal were selected for each section. Cell counting was then performed on the imaged sections. To present normalized data in units of DG (cells in DG), the cell number was calculated at the same surface area under magnification, and 4 to 6 brain sections per animal were selected. Throughout the analysis, we used ki67 as a marker of cell division, Nestin as a marker of NSCs, DCX as a marker of immature neurons, and NeuN as a marker of mature neurons. We identified type–1 NSCs as a radial GFAP-positive process and SOX2-positive nucleus or radial Nestin-positive cells in the subgranular zone (SGZ), and type–2 NSCs as non-radial Nestin-positive cells. We assessed changes in the number of NSCs at different time points (PND 1, PND 10 and PND 21) by counting the total number of Nestin-positive cells.

### 2.5. Western Blotting Analysis

Samples were lysed with radioimmunoprecipitation assay (RIPA) buffer (Beyotime, Shanghai, China) supplemented with a proteinase inhibitor cocktail (Roche, Mannheim, Germany) for 40 min on ice. Subsequently, lysates were harvested and centrifuged at 12,000× *g* rpm for 15 min at 4 °C. BCA protein assay kit (Beyotime Biotechnology, Shanghai, China) was used to determine the total protein concentration, according to the manufacturer’s instructions. For each run, 30 μg protein was mixed with the corresponding sodium dodecyl sulfate (SDS) gel sample buffer, boiled for 10 min, separated on 10% SDS-polyacrylamide gels, and transferred onto 0.45 μm polyvinylidene fluoride (PVDF) membranes. The membranes were blocked in 5% skim milk (BD, USA) at RT for 2 h and incubated with primary antibodies for total and phospho-Akt (1:1000, Cell Signaling Technology, Danvers, MA, USA) and GAPDH (1:10,000, Proteintech, Wuhan, China) overnight at 4 °C. The membranes were washed with Tris-buffered saline with 0.1% Tween (TBST) and subsequently reacted with the corresponding secondary antibodies (1:10,000, goat anti-rabbit IgG (H+L) HRP and goat anti-mouse IgG (H+L) HRP, Proteintech, Wuhan, China) for 1.5 h at RT. After washing with TBST, the protein bands were visualized using an enhanced chemiluminescence (ECL) detection kit (Millipore, MA, USA). GAPDH expression was used as an internal control. The optical densities of the bands on the films were quantified using Image J (version 1.61).

### 2.6. Statistical Analysis

All data were presented as mean ± standard error of the mean (SEM) and analyzed by GraphPad Prism (version 8.0; GraphPad Software, San Diego, CA, USA), and *p* < 0.05 was considered to statistical significance. The hippocampal neurogenesis on PND 1 and NSCs depletion rate between two groups were analyzed by Student’s *t* test (two-tailed unpaired). Other data including brain structure development, proliferation and differentiation of NSCs and expression of phospho-Akt at different time points were analyzed by 2-way ANOVA, followed by the Holm–Sidak’s multiple comparison test.

## 3. Results

### 3.1. Maternal HFD Alters Offspring Brain Structural Development

A maternal HFD animal model was constructed following standard procedure (Figure 1A). Our previous work showed that maternal HFD intake during pregnancy and lactation affects the metabolic phenotype of dams, including higher intake energy, increased adiposity, and higher plasma leptin and insulin concentrations [26,29]. In the current study, we focused on the effects of a maternal HFD on offspring brain development.

The results revealed that HFD-fed dams gained more weight than CHD-fed dams from gestation day 10 until parturition (*p* < 0.0001) (Figure 1B). There was no difference in litter size and offspring body weight on PND 1. However, from PND 10 to PND 21, offspring body weight dramatically increased in the HFD group (*p* < 0.0001) (Figure 1C,D), supported by previous observations [26]. Consistent with alterations in offspring body weight, the BPD and OFD of pups’ brain increased (*p* < 0.0001) in the HFD group on PND 10 and PND 21 (Figure 1F–H). No difference was noted between the two groups in brain wet weight of offspring during early postnatal development (Figure 1E). No significant difference was noted between HFD and CHD in the neocortical development of the offspring either (Figure 1I,J). Unlike in the CHD group, in which the thickness of GCL continued to increase from PND 1 to PND 21, the pups in the HFD group showed a faster increase from PND 1 (*p* = 0.006) to PND 10 and remained unchanged from PND 10 to PND 21 (*p* = 0.0016) (Figure 1K,L).

### 3.2. Maternal HFD during Pregnancy Attenuates Neurogenesis of Offspring DG at Birth

In rats, the granular cells of DG genesis at embryonic day 20 (E 20) and the DG would be recognizable as a morphological entity at E21–E22 [30]. Therefore, the effects of maternal HFD on NSC pool size and DG neurogenesis at birth (Figure 2A) were investigated. The number of Nestin^+^ (*p* = 0.0019) (Figure 2B,C) and Nestin^+^/Ki67^+^ cells (Figure 2B,D) (*p* = 0.002) significantly decreased in the HFD group. Similarly, the number of DCX^+^ cells (*p* = 0.0096) (Figure 2E,G) and NeuN^+^ cells (*p* = 0.0039) (Figure 2F,H) decreased in the HFD group.

### 3.3. Maternal HFD Accelerates Depletion of Offspring NSCs in DG from Birth to Weaning

In order to test whether a maternal HFD-induced lower capacity for NSC division at birth accelerates NSCs depletion, we examined the changes of NSCs proliferation in DG from birth to weaning (PND 1, PND 10 and PND 21). As expected, the number of Nestin^+^ cells dramatically decreased in the HFD group (*p* < 0.0001) at all time points (Figure 3A,B). Furthermore, starting from PND 1, the number of Nestin^+^ cells underwent an immediate and rapid decline and almost reduced by half on PND 10 (*p* = 0.0018). The reduction then continued to PND 21 (*p* = 0.0006) in both groups. During this process, the rate of Nestin^+^ cell reduction differed significantly between the two groups at different time points. It was approximately 39% in the HFD group but 57% in the CHD group from PND 1 to PND 10 (*p* = 0.0055). The reduction rate reached 78% in the HFD group from PND 10 to PND 21, while this change was not observed in the CHD group (*p* = 0.0266) (Figure 3A,C). In other words, rapid depletion of NSCs in the HFD group started immediately after the peak of neurogenesis (PND 7) during hippocampal development. Similarly, the number of Nestin^+^/Ki67^+^ cells (dividing NSCs) in GCL of DG dramatically decreased in the HFD group at all time points (*p* < 0.0001) (Figure 3D,E), and maintained a lower rate of dividing ability on PND 1 and PND 10 compared to the CHD group (*p* = 0.008) (Figure 3D,F).

### 3.4. Maternal HFD Induces a Decrease in Type-2 NSCs

There are two types of NSCs in SGZ. Changes of two subtypes of NSCs in DG reflect variations during distinct stages of neurogenesis. Type–1 NSCs have astrocytic features and are marked by GFAP. They have been reported as early progenitors. Type–2 cells are generated from type–1 NSCs. Most type–2 NSCs are proliferative, as late progenitors, and are related to cell migration during the formation of granular cell layers [31]. Nestin and SOX2 proteins are expressed in type–1 NSCs, and their expression persists into the type–2 NSC stage [14]. Here, we noticed that type–1 NSCs were characterized by their long processes toward the molecular layer (Figure 3A,D and Figure 4A), whereas most type–2 NSCs did not display long processes (Figure 2B). Regarding the type–1 NSCs, the number of radial Nestin^+^ cells (Figure 4A,B) and GFAP^+^/SOX2^+^ cells (Figure 4A,D) did not differ significantly at PND 1 or PND 10 between two groups, and they both decreased on PND 21 (*p* = 0.0096; *p* = 0.0184) (Figure 4A,B,D,E). Unlike type–1 NSCs, the number of type–2 NSCs (non-radial Nestin^+^ cells/SOX2^+^/GFAP^−^ cells) continually decreased from PND 1 to PND 21 (*p* < 0.0001). The percentage of type–2 NSCs was significantly less in the HFD group (Figure 4A,C,E).

### 3.5. Maternal HFD Reduces Offspring NSCs Stemness Maintenance in DG

NSCs self-renewal and neuronal differentiation require SOX2-dependent regulation during DG development [32,33]. We assessed the expression of SOX2 and its regulator Akt phosphorylation. The number of SOX2^+^ cells maintains the trend of a continuous decrease from birth to weaning in both groups (*p* < 0.0001). The number of SOX2^+^ cells dramatically decreased during development and was significantly less in the HFD group on PND 1 (*p* = 0.0007) and PND 21 (*p* = 0.0063). However, no significant difference was observed on PND 10 between the two groups (Figure 5A,B). In addition, the level of Akt phosphorylation in the offspring hippocampus in the HFD group was significantly lower than that in CHD group at all time points (*p* = 0.0023 on PND 1, *p* = 0.027 on PND 10 and *p* = 0.0163 on PND 21, Figure 5C,D).

### 3.6. Maternal HFD Speeds Up Neuronal Differentiation in DG

Approximately 85% of neurons within the DG are produced after birth, and the newly generated neurons are mainly involved in the structural formation of the DG [34]. To gain insight into the effects of maternal HFD on neuronal generation and maturation during offspring DG development, we assessed changes in the number of immature (DCX^+^) and mature neurons (NeuN^+^) in DG on PND 1, PND 10 and PND 21. The results showed that the number of DCX^+^ cells in the HFD group was significantly less than that in CHD group (*p* = 0.0103 on PND 1, *p* = 0.001 on PND 10 and *p* < 0.0001 on PND 21; Figure 3J and Figure 6A,C). Conversely, the number of NeuN^+^ cells in the HFD group altered differently from that in the CHD group, showing that the number of NeuN^+^ cells dramatically increased from PND 1 (*p* = 0.0169) to PND 10 (*p* = 0.0303) but decreased from PND 10 to PND 21 (*p* = 0.0034) (Figure 3K and Figure 6B,D). Furthermore, more than 50% of the DCX^+^ cell soma was found in the middle of GCL in the HFD group but not in the CHD group.

## 4. Discussion

In the current study, maternal HFD during pregnancy did not affect the litter size or offspring body weight and brain structures, including neocortical thickness, BDP and OFD on PND 1. However, the offspring gained more body weight in the HFD group on PND 10 and PND 21, which was consistent with previous observations [26]. Previous research indicated that maternal HFD leads to an increase in offspring blood glucose, plasma leptin and body fat and usually accompanies increasing body weight [6]. More evidence from human epidemiological and rodent experimental studies demonstrated that maternal HFD is associated with neurodevelopment disorders in the offspring [3,35,36]. Our results showed that when maternal HFD continued during lactation, the BPD and OFD of offspring became wider. The development of the GCL was delayed, showing a thinner GCL on PND 1 and PND 21 in the HFD group. This indicates that maternal HFD during lactation increased brain size but decreased GCL thickness in the hippocampus of the offspring.

NSCs proliferation, migration, timing and dynamic cell–cell interactions form the basis of DG formation during the early postnatal period [37]. The immediate reduction in the number and in proliferation caused by early postnatal insults contributed to long-term deficits in DG neurogenesis [38]. A previous study demonstrated that maternal HFD during pregnancy and lactation affects offspring hippocampal neurogenesis in adulthood [22]. Our results revealed that maternal HFD during pregnancy and lactation attenuates hippocampal development, particular DG development during the early postnatal stage. Maternal HFD during pregnancy induced a dramatic decrease in NSCs amount and their proliferation (Nestin^+^/Ki67^+^), differentiation (DCX^+^, NeuN^+^), and stemness maintenance (SOX2^+^) in DG of offspring, leading to a decline of the pool size of NSCs at birth. The size and properties of the NSC pool determine the capacity of adult neurogenesis [39,40]. Therefore, we inferred that maternal HFD intake during pregnancy may also affect hippocampal development neurogenesis.

Previous findings have proven that the transition of neurogenesis from a high level to a lower level during early postnatal DG development helps to maintain the stemness of stem cells throughout adult life [13]. Our results indicated that maternal HFD intake during lactation disturbed this transition. In the current study, maternal HFD during lactation accelerated the depletion of NSCs pool, with a significant reduction in the number and proliferation of NSCs at different time points (PND 1, PND 10, and PND 21). Meanwhile, the rate of NSCs depletion was lower from PND 1 to PND 10 and higher from PND 10 to PND 21. Nevertheless, there was no significant difference between groups in the proportion of proliferating NSCs on PND 21, although the number of proliferating NSCs decreased.

Symmetric self-renewal and the consumption of symmetric differentiation divisions can reflect the level of neurogenesis throughout life [41,42]. In the NSCs population, a small fraction of type–1 NSCs symmetrically divide into self-renewal cells, and 70% to 80% of type–1 NSCs are consumed by the asymmetric generation of type–2 cells. This results in the depletion of type-1 cells over time [41,43,44]. We noticed that maternal HFD induced a dramatic reduction in type–2 NSCs (non-radial Nestin^+^) during DG development. This is consistent with the decreased number of NSCs and proliferating NSCs. In addition, there was a decrease in type–1 NSCs (SOX2^+^/GFAP^+^) in the HFD groups on PND 21, but there was no significant difference between two groups on PND 1 and PND 10. The present data suggest that type–2 NSCs are the main cell types involved in proliferation during early postnatal DG development and that maternal HFD interrupted this behavior.

The phosphatidylinositide 3 kinases (PI3Ks)/Akt signaling pathway regulates cell proliferation, survival and metabolic processes [45]. Some findings suggest that PI3K/Akt signaling controls stem cell pluripotency by preserving the self-renewal and differentiation abilities in an SOX2 activation-dependent manner [46,47,48]. The lower number of SOX2^+^ NSCs and decreased level of Akt phosphorylation in the offspring hippocampus at different time points may be due to maternal HFD. Our results showed that maternal HFD reduced the stemness maintenance of NSCs, and SOX2 expression was not entirely dependent on Akt phosphorylation during early postnatal DG development. Further experimentation is required to investigate the potential relationship between Akt activity and SOX2 expression after maternal HFD intake. A previous study showed that the asymmetric SOX2^+^ NSCs were type–2 NSCs. The asymmetric division of type–2 NSCs plays an important role in maintaining the size of the NSC pool [49]. Consistent with these, we found that maternal HFD reduced the NSCs pool size and decreased type–2 NSCs numbers, which was accompanied by a reduction in SOX2^+^ cell numbers. Additionally, SOX2 is not essential for NSC proliferation, although it is important for self-renewal [48]. This could explain why the number of NSCs and proliferating NSCs decreased dramatically in the HFD group on PND 10, but the SOX2 expression level did not change in the CHD group.

Furthermore, we investigated the effects of maternal HFD on neuronal genesis. It has been confirmed that the number of neurons in the rat hippocampus increased significantly during the first postnatal week and then decreased by 70% during the second postnatal week. After two weeks, the number of neurons increased dramatically again [50]. A decrease or loss of asymmetric cell division can be associated with reduced differentiation during DG development [15,51,52]. Our results showed that the number of immature neurons (DCX^+^) decreased in the HFD group at different time points (PND 1, PND 10 and PND 21). However, the number of mature neurons (NeuN^+^) did not change. The number of mature neurons increased from PND 1 to PND 10 but decreased from PND 10 to PND 21 in HFD group. This is contrary to the changes in proper neuronal development.

In the current study, we did not investigate the effects of maternal HFD on neuronal survival or morphological alterations. Therefore, whether maternal HFD will cause abnormal neuronal development and maturation during the early postnatal hippocampal development period is still unknown. In addition, the existing data showed that the response to HFD is sexually dimorphic [53,54]. In the current study, both male and female pups were involved, and the data were analyzed without gender selectivity. Further investigations are still required.

In conclusion, our results provide evidence that excessive fat intake during pregnancy and lactation could impair offspring DG development in the early postnatal development period by altering the division and differentiation of type–2 NSCs. This might lead to delayed hippocampal neurogenesis and cognitive disorders in adult offspring.

## Figures and Tables

**Figure 1 nutrients-14-02813-f001:**
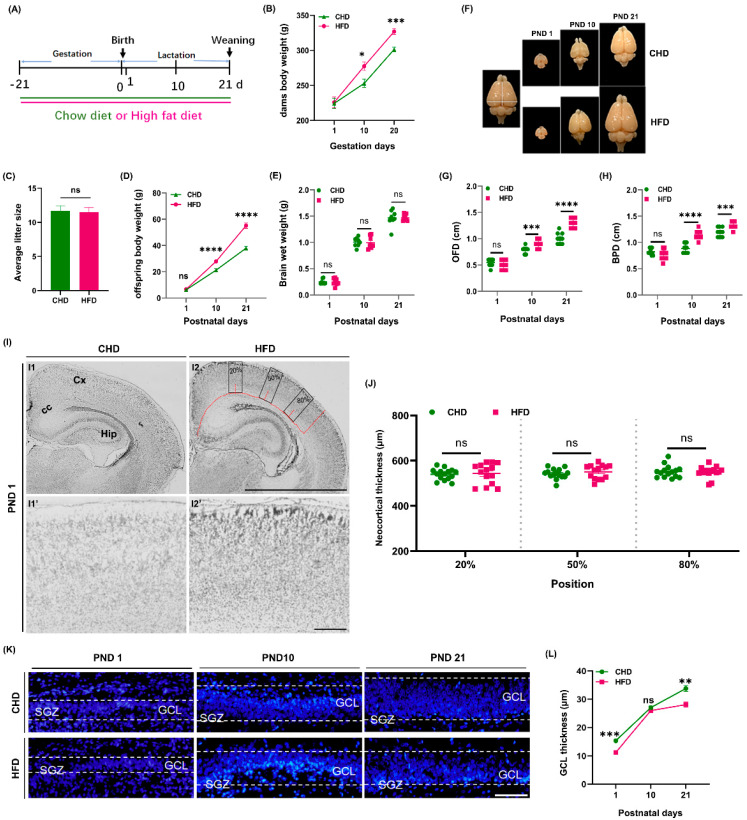
Maternal HFD affected offspring brain structure during development. (**A**) Schematic diagram of pregnant rats provided with CHD or HFD. (**B**) Changes in maternal body weight during gestation (*n* = 6) (**C**) Average litter size (*n* = 6). (**D**) Changes in offspring body weight during lactation (*n* = 18). (**E**) Measurement of brain wet weight on PND 1, PND 10 and PND 21 (*n* = 8). (**F**) Gross anatomy of brain of offspring on PND 1, PND 10 and PND 21. (**G**,**H**) Measurement of OFD (**G**) and BPD (**H**) of offspring on PND 1, PND 10 and PND 21 (*n* = 11). (**I**) Cortical morphology at different brain regions on PND 1 in rat offspring; scale bar = 20 μm. (**J**) Measurement of neocortical thickness (*n* = 4). (**K**) Changes in GCL thickness on PND 1, PND 10 and PND 21; scale bar = 50 μm. (**L**) Measurement of GCL thickness (*n* = 7). * *p* < 0.05, ** *p* < 0.01, *** *p* < 0.001, **** *p* < 0.0001 vs. CHD. ns, not significant; BPD, biparietal diameter; OFD, occipitofrontal diameter; Cx, cortex; CC, corpus callosum; Hip, hippocampus; SGZ, subgranular zone; GCL, granular cell layer.

**Figure 2 nutrients-14-02813-f002:**
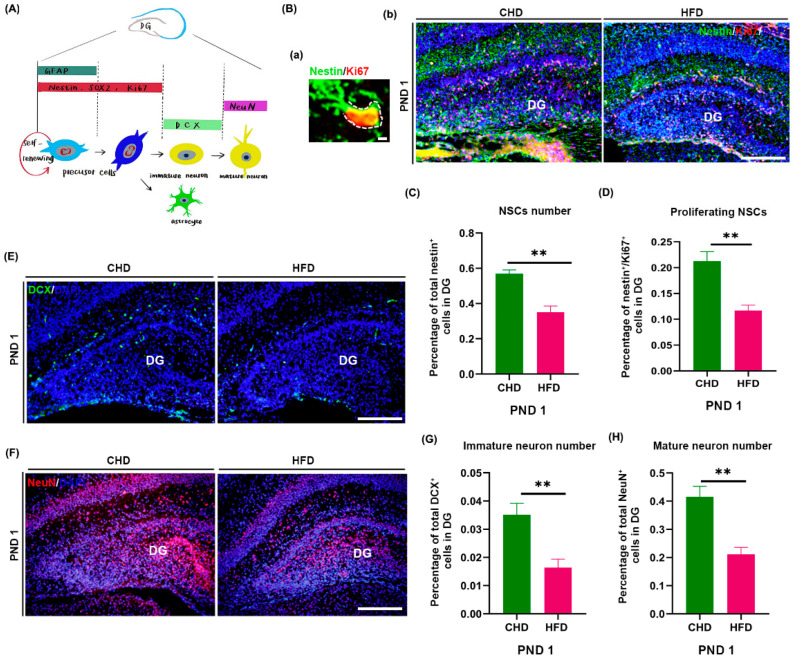
Maternal HFD reduced proliferation and neuronal differentiation of NSCs in offspring DG at birth. (**A**) Overview of neurogenesis originating from a sequence of NSCs in the developmental hippocampus. (**B**) Proliferation of NSCs on PND 1, non-radial NSCs morphology (a), scale bar = 200 μm; Nestin (green)/Ki67 (red) immunostaining (b), scale bar = 50 μm. (**C**) Quantification of NSCs number (*n* = 4). (**D**) Quantification of proliferating NSCs number (*n* = 5). (**E**) DCX (green) immunostaining, scale bar = 50 μm. (**F**) NeuN (red) immunostaining, scale bar = 50 μm (*n* = 4). (**G**) Quantification of immature neurons number (*n* = 4). (**H**) Quantification of mature neurons number (*n* = 4). ** *p* < 0.01 vs. CHD. DG, dentate gyrus; NSCs, neural stem cells; DCX, doublecortin; PND, postnatal day.

**Figure 3 nutrients-14-02813-f003:**
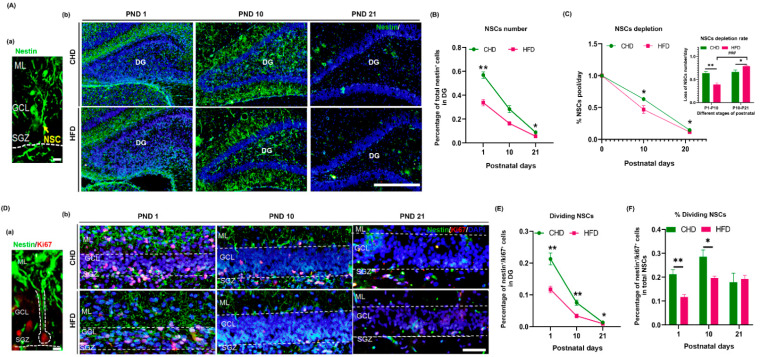
Maternal HFD reduced NSC proliferation during DG development. (**A**) Changes in NSC number on PND 1, PND 10 and PND 21; morphology of radial NSCs (Nestin^+^) (a), scale bar = 200 μm; Nestin (green) immunostaining (b), scale bar = 50 μm. (**B**) Quantification of NSCs number on PND 1, PND 10 and PND 21 (*n* = 3). (**C**) The rate of NSC depletion on PND 1 (*n* = 3), PND 10 (*n* = 5) and PND 21 (*n* = 4). (**D**) Changes in proliferating NSC number on PND 1, PND 10 and PND 21; proliferation of radial NSCs morphology (Nestin^+^/Ki67^+^) (a), scale bar = 200 μm; Nestin (green)/Ki67 (red) immunostaining in GCL (b), scale bar = 50 μm. (**E**) Quantification of Ki67^+^ dividing NSCs on PND 1 (*n* = 3), PND 10 (*n* = 3) and PND 21 (*n* = 4). (**F**) Percentage of Ki67^+^ dividing NSCs out of total NSCs. * *p* < 0.05, ** *p* < 0.01, vs. CHD; ^###^
*p* < 0.001 vs. postnatal days. DG, dentate gyrus; NSCs, neural stem cells; ML, molecular layer; GCL, granular cell layer; SGZ, subventricular zone; PND, postnatal day.

**Figure 4 nutrients-14-02813-f004:**
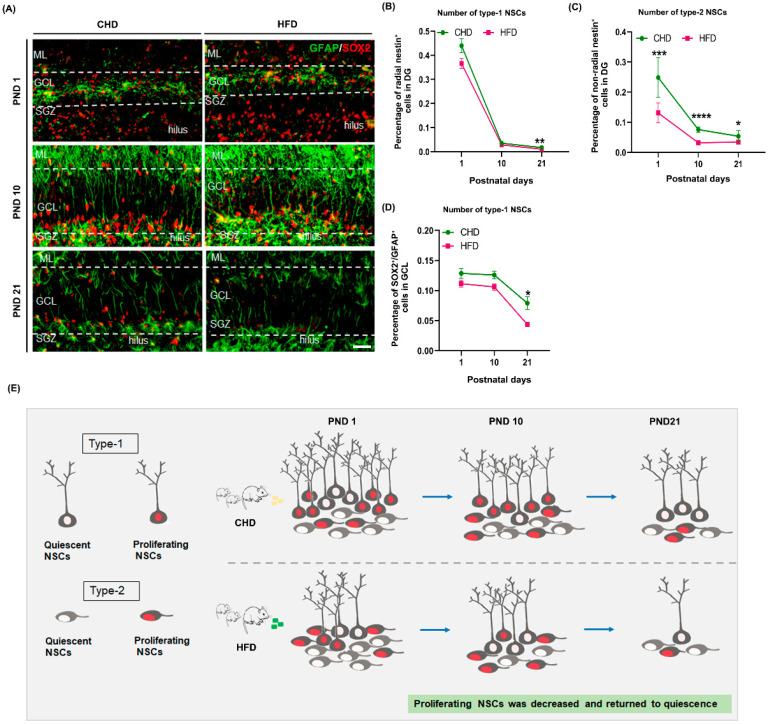
Maternal HFD leads to a decrease in type-2 NSCs during DG development. (**A**) GFAP (green)/SOX2 (red) immunostaining on PND 1, 10 and 21, scale bar = 50 μm. (**B**,**D**) Quantification of type–1 NSCs (radial Nestin^+^ cells and GFAP^+^/SOX2^+^ cells) on PND 1 (*n* = 5), 10 (*n* = 3) and 21 (*n* = 4). (**C**) Quantification of type–2 NSCs (non-radial Nestin^+^ cells) on PND 1 (*n* = 5), 10 (*n* = 3) and 21 (*n* = 4). (**E**) Summary of effects of maternal HFD on two subtypes of NSCs during DG development. * *p* < 0.05, ** *p* < 0.01, *** *p* < 0.001, **** *p* < 0.0001 vs. CHD. NSCs, neural stem cells; PND, postnatal day; ML, molecular layer; GCL, granular cell layer; SGZ, subventricular zone; GFAP, glial fibrillary acidic protein.

**Figure 5 nutrients-14-02813-f005:**
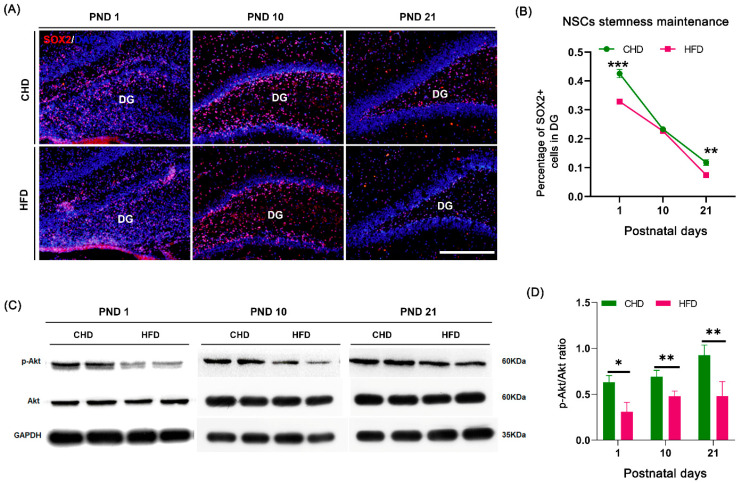
Maternal HFD affected NSC stemness maintenance during DG development. (**A**) SOX2 (red) immunostaining on PND 1, PND 10 and PND 21, scale bar = 50 μm. (**B**) Quantification of NSC stemness maintenance (SOX2^+^ cells) on PND 1, PND 10 and PND 21 (*n* = 3). (**C**) Western blotting of Akt and Akt phosphorylation in hippocampus on PND 1, PND 10 and PND 21. (**D**) Quantification of Akt phosphorylation to total Akt in hippocampus on PND 1, PND 10 and PND 21 (*n* = 4). * *p* < 0.05, ** *p* < 0.01, *** *p* < 0.001 vs. CHD. DG, dentate gyrus; PND, postnatal day.

**Figure 6 nutrients-14-02813-f006:**
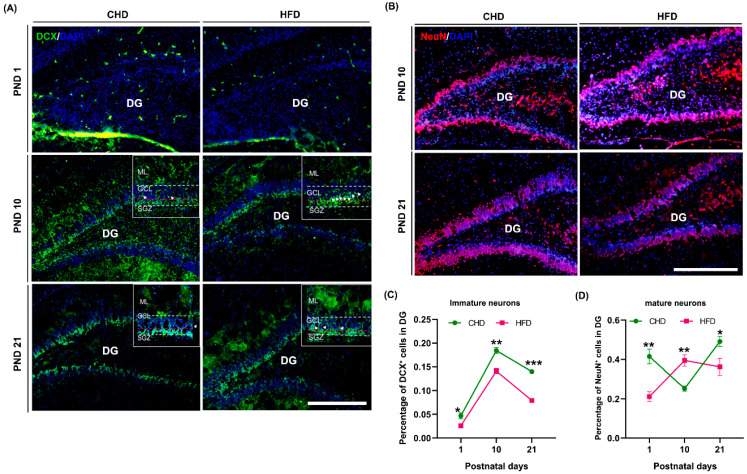
Maternal HFD affected neuronal generation during DG development. (**A**) DCX (green) immunostaining on PND 1, 10 and 21, scale bar = 50 μm. (**B**) NeuN (red) immunostaining on PND 1, PND 10 and PND 21, scale bar = 50 μm. (**C**) Quantification of immature neurons (DCX^+^ cells) on PND 1 (*n* = 3), PND 10 (*n* = 3) and PND 21 (*n* = 4). (**D**) Quantification of mature neurons (NeuN^+^ cells) on PND 1 (*n* = 3), PND 10 (*n* = 3) and PND 21 (*n* = 4). * *p* < 0.05, ** *p* < 0.01, *** *p* < 0.001 vs. CHD. DG, dentate gyrus; DCX, doublecortin; PND, postnatal day.

## Data Availability

The data presented in this study are available from the corresponding author.

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
