# Peer review of "Maternal High-Fat Diet Reduces Type-2 Neural Stem Cells and Promotes Premature Neuronal Differentiation during Early Postnatal Development"

_nutrients, 2022, doi:10.3390/nu14142813_

Round 1
Reviewer 1 Report
This study examined the effect of high-fat diet (HFD) on neurogenesis during early postnatal period. Results suggest that HFD reduced neuro stem cells and their proliferation. The research methods are solid and findings are interesting. Here are a few questions to consider:
1. Since there are 10 pups in each litter, how is the random factor of litter controlled for in statistical analysis, i.e., the pups in the same litter may be more similar to each other than pups in other litters. How is the sex of offspring controlled for in the analysis for potential sexually dimorphic responses to HFD.
2. The sentence in line 39-41 "The prevalence of..." is confusing, why is the prevalence of obesity a cause of the effect of HFD on neurogenesis?
Author Response
This study examined the effect of high-fat diet (HFD) on neurogenesis during early postnatal period. Results suggest that HFD reduced neuro stem cells and their proliferation. The research methods are solid and findings are interesting. Here are a few questions to consider:
We greatly appreciated your valuable comments. Following your suggestions, the corrections have been made carefully.
Q1. Since there are 10 pups in each litter, how is the random factor of litter controlled for in statistical analysis, i.e., the pups in the same litter may be more similar to each other than pups in other litters. How is the sex of offspring controlled for in the analysis for potential sexually dimorphic responses to HFD.
Response:We are very grateful for reviewer’s comments. In this study, we randomly selected the pups from three dams in each time point to avoid the bias. In statistical analysis, both male and female pups were concerned without gender selectivity. Some existed data showed that the maternal HFD may impair the learning and memory of male offspring via epigenetic regulation [1-2]. This indicated that there are sexually dimorphic responses to HFD. We appreciated very much for reviewer’s suggestion. The limitation of our data has been clarified in the revised manuscript (line 396-399).
- Robb, J.L.; Messa, I.; Lui, E.; Yeung, D.; Thacker, J.; Satvat, E.; Mielke, J.G. A maternal diet high in saturated fat impairs offspring hippocampal function in a sex-specific manner. Behav Brain Res 2017, 326, 187-199, doi:10.1016/j.bbr.2017.02.049.
- Glendining, K.A.; Higgins, M.B.A.; Fisher, L.C.; Jasoni, C.L. Maternal obesity modulates sexually dimorphic epigenetic regulation and expression of leptin receptor in offspring hippocampus. Brain Behav Immun 2020, doi:10.1016/j.bbi.2020.03.006.
Q2. The sentence in line 39-41 "The prevalence of..." is confusing, why is the prevalence of obesity a cause of the effect of HFD on neurogenesis?
Response:We are very sorry for the confusing statement. Here we wish to say that the risk of neurodevelopmental disorders in offspring increased with the rise of maternal obesity. Maternal HFD is one of the main causes of maternal obesity and diabetes which were involved in offspring health and the programming of neural development. The confusing sentence has been modified and highlighted in the revised manuscript (lines 38-40).
Reviewer 2 Report
Hu et al. investigated how excessive fat intake in pregnant female rats affects NSCs behavior during the early DG development of offspring.. Overall, this study is well conducted and carries novel findings.
Below are my comments:
Conclusion needs to be modified. The authors stated “may impair adult hippocampal neurogenesis” (line 33), however, this study examined
PND 21 young rats.
Resolution of Fig 2A is not good.
Please specify what results were analyzed by 2-Way ANOVA.
The authors should mention the limitation of no sex discrimination.
Reference styles were inconsistent, with irregular appearance of upper case letter at word start on cited works.
Some typos: line 41, high-fat diet, be HGD; line 181, be HFD and etc; line 182, please delete “/”;
At line 25, “NSCs behaviors, in terms of proliferation and differentiation, immunostaining with Nestin, Ki67, SOX2, Doublecortin, and NeuN were applied and the number of positive cells was calculated.” This sentence looks weird to me.
At line 323, “Previous research indicates that maternal HFD leads to an increase in blood glucose, plasma leptin, and body fat, and this change is accompanied by increased body weight [31,32].” This sentence looks weird to me. Dam or offspring?
Author Response
Reviewer 2:
Hu et al. investigated how excessive fat intake in pregnant female rats affects NSCs behavior during the early DG development of offspring. Overall, this study is well conducted and carries novel findings.
Below are my comments:
We are very grateful for reviewer’s comments. According to the suggestions, the point-by-point responses and revisions have been made carefully.
Q1. Conclusion needs to be modified. The authors stated “may impair adult hippocampal neurogenesis” (line 33), however, this study examined PND 21 young rats.
Response:We deeply appreciate reviewer’s suggestion. In the current study, we wish to say that the maternal HFD reduces the pool size and capability of NSCs. This may impair the potential of hippocampal neural regeneration in adult hood. We agree with reviewer’s comment that our conclusion need to be modified, since we did not examine the pups’ hippocampal NSCs when they grow up to be adult. The revision has been made and highlighted in the revised manuscript (lines 31-33).
Q2. Resolution of Fig 2A is not good.
Response:We are very grateful for reviewer’s comments. The new Fig.2A with high resolution has been added to figure 2 to replace the previous one.
Q3. Please specify what results were analyzed by 2-Way ANOVA.
Response: We are sorry for the muddy information. Following reviewer’s suggestion, the data that were analyzed by 2-way ANOVA has been specified. Corrections have been made and highlighted in the revised manuscript (line 166-170).
Q4. The authors should mention the limitation of no sex discrimination.
Response: We are very grateful for reviewer’s suggestion. In the current study, both male and female pups from different dams were randomly selected and the data were analyzed without gender selectivity. We agree with reviewer’s comment that the response to HFD is sexually dimorphic [1-2]. Following reviewer’s suggestion, the limitation of current work has been mentioned clearly in the revised manuscript (line 396-399).
- Robb, J.L.; Messa, I.; Lui, E.; Yeung, D.; Thacker, J.; Satvat, E.; Mielke, J.G. A maternal diet high in saturated fat impairs offspring hippocampal function in a sex-specific manner. Behav Brain Res 2017, 326, 187-199, doi:10.1016/j.bbr.2017.02.049.
- Glendining, K.A.; Higgins, M.B.A.; Fisher, L.C.; Jasoni, C.L. Maternal obesity modulates sexually dimorphic epigenetic regulation and expression of leptin receptor in offspring hippocampus. Brain Behav Immun 2020, doi:10.1016/j.bbi.2020.03.006.
Q5. Reference styles were inconsistent, with irregular appearance of upper case letter at word start on cited works.
Response: We apologize for the inconsistent reference style. Following reviewer’s suggestions, all references have been carefully double checked and the related corrections, including the capitalized letter used at word start on cited works, the abbreviation of the journal title and the information of DOI have been made in the revised manuscript (Page 13-16).
Q6. Some typos: line 41, high-fat diet, be HGD; line 181, be HFD and etc; line 182, please delete “/”;
Response: We sincerely apologize for the typos. The entire manuscript has been checked carefully for the grammatical and typing errors. The “/” on line 174 has been deleted and the other mistakes had been corrected and highlighted in the revised manuscript.
Q7. At line 25, “NSCs behaviors, in terms of proliferation and differentiation, immunostaining with Nestin, Ki67, SOX2, Doublecortin, and NeuN were applied and the number of positive cells was calculated.” This sentence looks weird to me.
Response: We apologize for the vague description. This sentence has been re-written as “Hippocampal NSCs behaviors, in terms of proliferation and differentiation, were investigated after immunohistochemical staining with Nestin, Ki67, SOX2, Doublecortin (DCX) and NeuN” (line 25-27).
Q8. At line 323, “Previous research indicates that maternal HFD leads to an increase in blood glucose, plasma leptin, and body fat, and this change is accompanied by increased body weight [31,32].” This sentence looks weird to me. Dam or offspring?
Response: We are very sorry for the confusing statement. Here we wish to say the offspring’s changes. Following reviewer’s suggestion, this sentence has been modified as “Previous research indicated that maternal HFD leads to an increase in offspring’s blood glucose, plasma leptin and body fat and usually accompanied with increasing of body weight” (line 319-321).